# Quantitative Proteomics of *Chromochloris zofingiensis* Reveals the Key Proteins Involved in Cell Growth and Bioactive Compound Biosynthesis

**DOI:** 10.3390/plants11141851

**Published:** 2022-07-15

**Authors:** Wen Qiu, Rongfeng Chen, Xianxian Wang, Junying Liu, Weiguang Lv

**Affiliations:** 1Eco-Environmental Protection Research Institute, Shanghai Academy of Agricultural Sciences, Shanghai 201403, China; qwyuyi@foxmail.com (W.Q.); lwei1217@sina.com (W.L.); 2Shanghai Engineering Research Center of Low-Carbon Agriculture, Shanghai 201403, China; 3Shanghai Agricultural Environment and Farmland Conservation Experiment Station of Ministry of Agriculture, Shanghai 201403, China; 4National Agricultural Experimental Station for Agricultural Environment, Fengxian, Shanghai 201403, China; 5National Center for Occupational Safety and Health, NHC, Beijing 102308, China; 18760606336@139.com; 6Institute of Toxicology and Genetics, Karlsruhe Institute of Technology, 76344 Eggenstein-Leopoldshafen, Germany; xianxian.wang@kit.edu; 7NatPro Center, School of Pharmacy and Pharmaceutical Sciences, Trinity College Dublin, D02 PN40 Dublin, Ireland

**Keywords:** microalgae, proteomics, growth, glucose

## Abstract

Glucose metabolism regulates cell growth and affects astaxanthin accumulation in the green algae *Chromochloris zofingiensis*. Hub gene functioning in this bioactive compound has been illustrated at the genome, transcriptome and metabolome level, but is rather limited from a proteome aspect. Microalgal cell produce an enhanced biomass (8-fold higher) but decreased lipid and astaxanthin content (~20% less) in the glucose condition compared to the control. Here, we investigate the proteomic response of *C. zofingiensis* grown with and without glucose using an LC-MS/MS-based Tandem Mass Tag (TMT) approach. The proteomic analysis demonstrated that glucose supplementation triggers the upregulation of 105 proteins and downregulation of 151 proteins. Thus, the carbon and energy flux might flow to cell growth, which increased the associated protein abundance, including DNA polymerase, translation initiation factor, 26S proteasome regulatory subunits, and the marker enzyme of the TCA cycle ribosomal protein. Moreover, the glucose supplement triggered the downregulation of proteins mainly involved in photosynthesis, chloroplasts, valine, leucine and isoleucine biosynthesis, 2-oxocarboxylic acid metabolism, and pantothenate and CoA biosynthesis pathways. This proteomic analysis is likely to provide new insights into algal growth and lipid or astaxanthin accumulation upon glucose supplementation, providing a foundation for further development of *C. zofingiensis* as oleaginous microalga for bioengineering applications.

## 1. Introduction

Natural bioactive compound production has attracted community interest because of the large impending demand of biopharmaceutical and human nutraceutical products [1]. Natural astaxanthin has antioxidant and anti-inflammatory traits, which benefits human disease treatments against cancer, cardiovascular disease, inflammatory disease, neurodegenerative disease, diabetes, and obesity [2]. *Chromochloris zofingiensis* and *Haematococcus pluvialis* represent the most promising candidates for natural astaxanthin production [3]. *C. zofingiensis* grows fast and could build up an ultrahigh cell density under heterotrophic conditions, as well as has a high tolerance toward contamination and unfavorable environments [4], making *C. zofingiensis* a good candidate for mass production of astaxanthin. In contrast, although accumulating a high content of astaxanthin, wide application of *H. pluvialis* is limited by low biomass yield, a slow growth rate, a high requirement of light for astaxanthin biosynthesis, and a high risk of contamination at the green stage [5]. However, astaxanthin yields from *C. zofingiensis* are too low to support industrial-scale commercial production for nutraceutical application, and development using biotechnology has high costs [4]. Therefore, cultivation and harvesting optimization, strain improvement, coproducts development to curtail the market price (which is particularly important), as well as understanding the nature of astaxanthin biosynthesis and the related complex regulatory mechanisms in *C. zofingiensis* are essential. 

Nutrients such as glucose are critical factors affecting microalgae growth, lipid metabolism, and astaxanthin accumulation [6]. Glucose supplementation causing high biomass accumulation has been reported for *C. zofingiensis* [7] and *Chlorella vulgaris* [8]. Feeding with 30 g/L glucose induces a higher (3.8-fold) astaxanthin production for *C. zofingiensis* ATCC30412 [9]. However, a controversial result was shown, where a 5 g/L glucose supplementation induced unstable astaxanthin production in *C. zofingiensis* ATCC30412: higher within 48 h, but lower (0.78-fold) at 96 h than the control without glucose [10]. 

Development of techniques enable researchers to exploit the key genes regulating astaxanthin production [9]. It also accelerates elucidating the mechanisms modulating lipid/astaxanthin accumulation in response to external stimuli [11]. However, the -omics response to astaxanthin biosynthesis of *C. zofingiensis* mainly has been examined at the genome, transcriptome, or metabolome level [9,12], whereas proteomics studies are rather limited. In this study, the reduction in astaxanthin accumulation (0.79-fold) could be offset by the extremely high microalgal biomass production (8-fold) with glucose addition. Further proteomic response of *C. zofingiensis* to glucose supplementation was investigated by a TMT-label experiment. Detailed analysis would provide a foundation for further development of *C. zofingiensis* as oleaginous microalga for bioengineering applications. 

## 2. Results

### 2.1. Cell Growth, Biomass, and Bioproduct Production of C. zofingiensis 

Compared to the control, glucose supplementation enhanced *C. zofingiensis* SAG 211-14 cell growth, from which a large amount of biomass (cell dry weight) was produced and accumulated through the consumption of glucose, therefore leading to heterotrophic growth. At Day 10, the treatment with glucose resulted in 0.52 g microalgal dry weight (calculated based on standard curve), which is 8-fold higher than the treatment without glucose (Figure 1a). An approximately 0.84- and 0.79-fold lower lipid content and astaxanthin accumulation, respectively, were observed in the presence of glucose (Figure 1b).

### 2.2. Proteomic Profile of C. zofingiensis Grown with/without Glucose

To have a good understanding of the differential protein expressions in response to glucose treatment, a TMT-based quantitative proteomics study was carried out. In total, 9915 unique sequences belonging to 1045 proteins were identified from the derived peptide-spectrum match (PSM) list, with the average PSM score higher than 145 when searching against the reference species (false discovery rate < 1%). About 90% of the peptides were distributed within the length of 8–25, with a peptide score higher than 150, indicating high data quality. Principal component analysis separated the proteins with/without glucose conditions clearly, for which PC1 explains 80.2% and PC2 explains 13% (Figure 2a).

Out of 957 proteins quantified for glucose vs. control for *C. zofingiensis*, 256 show significant abundance alterations, with a *p* < 0.05 and cutoff of 1.5-fold; 151 (15.8%) were downregulated and 105 (11.0%) were upregulated (Figure 2b). Hierarchical clustering analysis suggest that the differentially expressed proteins (DEPs) were well separated, which were associated with algal glucose response (Figure 2c). To illuminate the influence of glucose on *C. zofingiensis*, these DEPs were subjected to bioinformatics analysis. GO enrichment analysis showed DEPs are classified into photosynthesis, photosystem, chloroplast, plastid, and thylakoid-associated process (Figure 3a–c). The KEGG pathway enrichment analysis shows that the DEPs are assigned to the following categories (Figure 3d and Figure 4 and Appendix A): metabolic pathways (27.6%), biosynthesis of secondary metabolites (17.3%), biosynthesis of antibiotics (14.3%), biosynthesis of amino acids (10.2%), and carbon metabolism (9.2%). The DEPs were mainly enriched (*p* < 0.05) in valine, leucine and isoleucine biosynthesis, 2-oxocarboxylic acid metabolism, and the pantothenate and CoA biosynthesis pathway (Figure 4).

Proteins with increased abundance fell into several major biological processes. The presence of glucose triggers the increased abundance of proteins implicated in carbohydrate metabolism (Table 1), including succinate dehydrogenase, the marker enzyme of TCA cycle and UDP-glucose 6-dehydrogenase, as well as glutamine-fructose-6-phosphate transaminase, which is the first and rate-limiting enzyme of the hexosamine pathway and controls the flux of glucose into the hexosamine pathway, presumably to provide intermediates and energy for anabolism and hence algal growth. The downregulated proteins are acetohydroxyacid dehydratase, D-fructose-1,6-bisphosphate 1-phosphohydrolase (FBP1 and MNEG_7410), pyruvate phosphate dikinase, phosphopyruvate hydratase, and pyruvate dehydrogenase E1 component subunit alpha. Pyruvate carboxylase and sedoheptulose-1,7-bisphosphatase (SBPase), enzymes for the unique reactions in gluconeogenesis, were significantly repressed. SBPase is a rate-limiting enzyme in the Calvin cycle, which catalyzes the dephosphorylation of sedoheptulose-1,7-bisphosphate.

As for amino acid metabolism, six amino acid metabolism pathways with 23 DEPs were repressed/activated in the presence of glucose (Table 2). Of these, the hydrophobic branched chain amino acids (e.g., valine, leucine, and isoleucine) pathway was enriched. Furthermore, cysteine synthase A (OASTL3, MNEG_10819 and MNEG_11543) and 5-methyltetrahydropteroyltriglutamate-homocysteine S-methyltransferase (METE and MNEG_0007), involved in cysteine and methionine metabolism, were upregulated with glucose addition. 

Eleven of the downregulated proteins were involved in photosynthesis; five for photosystem I (psaA, psaB, psaC, psaD, psaN) and six for photosystem II (psbA, psbB, psbC, psbD, PsbP domain-containing protein, photosystem II stability/assembly factor), as listed in Table 3. Furthermore, subunits of ribulose bisphosphate carboxylase (rbcL, rbcL.3, rbcL.4, RBCS, and ribulose bisphosphate carboxylase/oxygenaseactivase) and thylakoid lumenal protein (chloroplastic 17.4 kDa protein and CPLD44) showed a significantly decreased abundance under glucose conditions. 

Moreover, the upregulation of diacylglycerol O-acyltransferase (DGAT1a) and acyl-carrier-protein desaturase (MNEG_9313) was identified, for which the former catalyzes the final step of the biosynthesis process of triacylglycerol (TAG) and the latter catalyzes the desaturation of the de novo synthesized fatty acids to different levels. We detected no-significant abundance variation in phosphoenolpyruvate carboxykinase, the activity of which is linked to CO_2_ fixation in algae. In addition, the proteins involved in energy production mostly were altered insignificantly; e.g., pyruvate kinase (PYK1, PYK2, MNEG_1868 and MNEG_11538), 2-oxoglutarate dehydrogenase (OGD1), isocitrate dehydrogenase (IDH1 and IDH3), and 6-phosphogluconate dehydrogenase (gnd) for NADPH production.

The protein–protein interaction (PPI) analysis results of the significant DEPs are shown in Figure 5 and Appendix A. A total of 112 nodes were searched and 59 proteins were classified into three clusters by k-means clustering using STRING; the growth-related proteins were mostly observed in Cluster 2 (green nodes in Figure 5). 

To further explore the underlying mechanism of the DEPs, five core modules were isolated from this PPI network using MCODE analysis (Figure 6a–e and Appendix A). There are 13 nodes in Module 1, including the proteins associated with photosynthesis (Appendix A). Module 2 contained 18 nodes, from which the algal growth-related proteins were mainly detected; for instance, the proteins related to transcription and translation (such as eukaryotic translation initiation factor 3, 5, 6 (EIF3I, MNEG_5528, EIF6A); 20S ribosome; 26S proteasome regulatory subunit 6; T1, T2, and RPT5; and T-complex proteins CCT1, CCT6, and CCT7), reflecting the significant change in *C. zofingiensis* in response to the glucose-induced conditions. In addition, expression of the major cell cycle—cell division-related proteins were coordinately regulated during glucose conditions (Figure 6b–e), including those characterized by a significant excess of upregulated proteins: eukaryotic initiation factor iso-4F, elongation factor Tu and EF-G, and adaptor protein complex 2 (AP2A1). To validate the protein–protein interaction, molecular docking analysis was carried out using ZDOCK. The Top 1 docking complex (cyan) indicates an interaction between the protein psbB (as receptor) and rbcL (as ligand), as visualized in Figure 6f. 

## 3. Discussion

### 3.1. Glucose Promote Rapid Growth of C. zofingiensis

Consistent with previous reports for *Chlorella (Chromochloris) zofingiensis* ATCC30412 [9,10] and E17 [7], *Chlorella protothecoides* [13], and *Chlorella vulgaris* [8], *C. zofingiensis* is able to accumulate a large amount of biomass under glucose-feeding conditions [7], while *C. vulgaris* can reach a high cell density during heterotrophic growth in fed-batch fermenters [8]. Here, we detected an 8-fold higher cell biomass of *C. zofingiensis* SAG 211-14 using 5 g/L glucose as carbon source. Moreover, the autotrophic condition of *C. reinhardtii* is characterized by the accumulation of lipids [14], while supplementing glucose recalibrates the *C. protothecoides* metabolism, leading to a constant decrease in cell lipid content after 72 h [13]. Fatty acid analysis using GC-MS revealed a decreased astaxanthin accumulation after glucose treatment for 96 h and 192 h (0.79- and 0.78-fold lower, respectively) for *C. protothecoides* [10,13]. In addition, it is consistent with results for the yeast *Phaffia rhodozyma*, where a low glucose concentration (10 g/L) enhances cell growth but inhibit astaxanthin production, while a high glucose concentration (20–80 g/L) shows the reverse effect [15]. However, it is contrasted against reports from Huang and co-authors, who detect the accumulation of 3.8-fold higher astaxanthin within 96 h of *C. zofingiensis* when fed with 30 g/L glucose [9], and Liu and colleagues, who observed the accumulation of astaxanthin [7].

### 3.2. Glucose Nutrient Induce Upregulation of Growth-Associated Protein of C. zofingiensis 

The proteomic study of *C. zofingiensis* in response to glucose treatment showed altered abundances of 957 proteins. The consumption of glucose requires the transport of sugar into algal cells, which is usually carried out by H+/hexose co-transporters [16]; therefore, we were not unexpecting to observe the upregulation of H+-transporting ATPase subunit A and H+ transporting ATP synthase; rather, the downregulation of ammonium transporter and Ca2+ transporter. 

Glucose prompted an abundant carbon skeleton, sustained microalgal rapid growth, and higher biomass generation, as well as induced a smaller cell size (unpublished data), promoted the significant upregulation of proteins related to 26S proteasome regulatory subunit (6, T1, T2 and RPT5) and AAA-ATPase of VPS4/SKD1 family (VPS4), eukaryotic translation initiation factor 5A (eIF5A), adaptor protein, and T-protein complexes of *C. zofingiensis* (Table 1). Consistent with previous reports of the biological function of these proteins in plants [17] or algae [12], results indicated their important role in diverse processes, including cell size regulation, cell expansion, cell proliferation rates, balancing and stress responses, as well as cell death programs [12,17]. Loss of function of the 26S proteasome subunit regulatory particle AAA ATPase causes an enlargement of shoot organ size in *Arabidopsis*, which compensates for the reduction in cell number [18]. EIF5A has been shown to be required for cell proliferation in yeast [19] and mammalian cells [20], and mutation of eIF5A in *Arabidopsis* results in phenotypical similarities characteristic of slow growth and defects in reproductive development [21]. Similarly, *Arabidopsis* adaptor protein complex 1 (AP-1) is required for cell division in and plant growth [22], while ap2m mutant plants exhibited delayed anther dehiscence and reduced stamen elongation, suggesting that AP-2 plays a role in floral organ development and plant reproduction [23]. Moreover, CCT2 and CCT3 silencing causes growth arrest in *Arabidopsis* with small round leaves [24]. 

### 3.3. Glucose Supplementation Decreases the Abundance of Proteins Involved in Amino Acid Metabolism 

Differentially expressed proteins were mainly enriched (*p* < 0.05) in valine, leucine and isoleucine biosynthesis, 2-oxocarboxylic acid metabolism, and the pantothenate and CoA biosynthesis pathway, which is consistent with the enrichment results of the transcriptome data for the microalga *Coccomyxa subellipsoidea* C-169 upon 2% CO_2_ vs. air supplementation [24]. Of these, the proteins showed reduced abundance and were enriched in the hydrophobic branched-chain amino acids (BCCAs) valine, leucine and isoleucine biosynthesis pathway: acetolactate synthase (ALSL1) catalyzes the first step [25], while aminotransferase (BCA2) catalyzes the final transamination reactions of the branched-chain amino acid biosynthesis [26]. BCCAs act as structural components of cell membranes modulating lipid homeostasis or turnover, and a concentration of as low as 0.1 mM of BCCAs shows a growth inhibitory role in the blue-green alga *Anabaena doliolum* [20]. BCA6 of *Arabidopsis thaliana* influence methionine chain elongation pathway [26], the regulation of flux into and through cysteine synthesis is of central importance for growth and fitness of microalgae because cysteine synthesis plays an integral role in the regulation of primary sulfur metabolism and the reaction intermediate of cysteine synthesis forms a direct connection with sulfate assimilation, carbon metabolism, and nitrate assimilation. Reversible redox post-translational modifications play an important role in regulation of cell metabolism by transforming cysteine residues into different forms [12].

### 3.4. Glucose Triggers the Downregulation of Proteins in Photosynthesis-Associated Processes

Glucose cultivation causes the downregulation of 11 proteins involved in photosynthesis (photosystem I and II, detailed in Appendix A), implying that photosynthesis and electron transport is largely inhibited in *Chlorella* cells with glucose supplement. It agrees well with the report for *Chlorella protothecoides* sp. 0710 [27] and *C. zofingiensis* SAG 211-14 [21,22], for which heterotrophic cultures with glucose were seen to have almost complete degradation of the enzymes associated with photosynthesis [13]. It also consistent with transcriptome results from Roth et al. [21,22], who observed the absence of photosynthetic activity and the significant decreased expressions of the genes for photosystem I and II in *C. zofingiensis* SAG 211-14 with the presence of glucose through RNA-seq. Indeed, the availability of glucose stops the necessity of algae to obtain organic carbon through the photosynthesis process [27] and triggers the turning-off of photosynthesis, degrading the photosynthetic apparatus and reducing the thylakoid membranes under light conditions [22]. On the other hand, glucose may alter the algal cellular lipid composition or cell structure; e.g., a decrease in chlorophyll content and chloroplast degradation where the astaxanthin biosynthesis process occurs [10]. 

Moreover, we detected the reduced abundance of other photosynthesis-associated proteins, including phosphoribulokinase, acetyl-CoA carboxylase, and pyruvate dehydrogenase complex SBPase in *C. zofingiensis* SAG 211-14. It was unexpected, because feeding glucose may recalibrate cell metabolism towards downstream intermediates and lipid accumulation in *C. protothecoides* [13]: phosphoribulokinase catalyzes the production of ribulose 1,5-bisphosphate, and the substrate functions to capture CO_2_ in photosynthesis and fatty acid synthesis in photosynthetic organisms [23]. As for acetyl-CoA carboxylase, it catalyzes the first and rate-limiting step for the fatty acid synthesis pathway, while the pyruvate dehydrogenase complex is involved in acetyl-CoA formation. Overexpression of SBPase in plants [24], microalgae *Dunaliella bardawil* [28], and *C. reinhardtii* [29] leads to a significant increase in photosynthesis, addressing the importance of these proteins. 

Furthermore, glucose supplementation might reduce the carotenoid biosynthesis sites because of smaller chloroplasts under glucose conditions (smaller cell size) and thus account for the lower astaxanthin content [10]. Glucose may directly regulate astaxanthin accumulation through modulating the expression of the key genes modulating astaxanthin biosynthesis, such as the β-carotenoid ketolase (*BKT*) and β-carotenoid hydroxylase (*CHYb*) genes of *C. zofingiensis* [30]. It may affect the transcription of BKT and CHYb genes through affecting de novo protein synthesis [30], because increasing the glucose supply decreased the protein content [15]. Although the key proteins involved in astaxanthin production (BKT and CHYb) were not detected, the inhibition response of fatty acid production with glucose supplementation (Figure 2) and the downregulation of photosynthesis-associated proteins (Table 3) verified the proposed linkage between photosynthesis and lipid production in the microalga *Eutreptiella* sp. [31]. In addition, glucose exposure enhanced the expression of the PDAT gene, which may promote the chloroplast decomposition, and thus reduce the available synthetic sites for astaxanthin biosynthesis in *C. zofingiensis* [10]. The balance between astaxanthin production, cell growth, and biomass accumulation could be modulated by supplementation with glucose (C/N ratio) for large-scale cultivation [15,21]. Accordingly, this study provides hints for its biotechnological modification: carbon sources such as glucose are used to provide more energy for higher growth rate as well as for respiration, and thus the cellular physiology and morphology would change due to the metabolic pathways of carbon assimilation and allocation being affected [27].

The network showed the expression of transcription- and translation-related proteins, reflecting the significant change in *C. zofingiensis* in response to the glucose-induced condition [32]. Altered-abundance proteins are likely to provide new insights into lipid accumulation in microalgal cells after glucose supplementation. Much work remains to gain a better understanding of the differences in regulation of the chloroplast structure and carbon flow upon glucose supply in algal cultures [33].

## 4. Material and Methods

### 4.1. Microalgal Species, Growth Media, and Culture Conditions

*C. zofingiensis* SAG 211-14, purchased from Germany, was cultured under photoautotrophic condition in BG11 medium with slight modification. The BG11 medium stock was obtained from CCAP, UK, and was diluted into the growth medium accordingly. A seed culture of *C. zofingiensis* was inoculated into a 50 mL Erlenmeyer flask from slant medium and grown at 25 °C, in a 16/8 h light/dark cycle, with a light intensity 30 µE m^−2^ s^−1^. After 20 days of nursery cultivation, the seed culture was transferred to a 250 mL Erlenmeyer flask to grow as a nursey inoculum under the same conditions. Then, 10 mL of culture (OD750 = 1.0 + 0.05) was inoculated into the growth BG11 medium with glucose (5 g L^−1^) [10], and no glucose addition was used as the control. The initial OD_750_ was adjusted similarly for the two algal inoculants. Samples were collected and measured at regular intervals to monitor their growth dynamics. Samples were harvested at 10 days in the growth curve to measure the lipid content and astaxanthin content with proteomics analysis conducted at the late phase. Biological triplicates were applied for each treatment.

### 4.2. Measurement of Dry Weight, Total Lipid Content, and Astaxanthin Content

Algae correlation analysis between the optical density (OD_750_) and dry weight was performed according to [34]. To determine the dry weight accurately, a set of correlation equations between the biomass and optical density was obtained by linear regression. Consequently, biomass can be calculated using the correlation equations by measuring OD_750_. The lipid content was measured using gravimetric methods with slight modifications [34]. Briefly, 500 µL of chloroform/methanol (2:1, *v*/*v*) were added to lyophilized algal cells and then sonicated for 1 min on ice. The supernatant was collected by centrifugation (3000× *g*, 10 min). The collected sample was adjusted for chloroform, methanol, and NaCl (2:1:1, *v*/*v*). The mixture was then centrifuged to separate the organic phase. The chloroform layer was collected and dried in a fume hood to a constant weight. The total lipid content was then calculated gravimetrically. Astaxanthin extraction was conducted as described by [35]. Briefly, 50 mg lyophilized algal cells were ground under liquid nitrogen and then 2 mL of acetic acid in DMSO was added and incubated at 70 °C for 5 min. The broken cells were extracted three times and centrifuged (5000× *g*, 3 min, 4 °C). Supernatants were collected and the absorbance was measured by a UV-spectrometer at 492 nm (A) [36]. The astaxanthin content was calculated based on the following equation: Astaxanthin content (%) = (A·25·dilution/2100·0.5 g) ·80%.

### 4.3. Protein Extraction and Quantification

Algal cultures (100 mL) were harvested and centrifuged at 3000× *g* for 10 min at 4 °C, and biological triplicates were applied. The pellets were washed twice using ddH_2_O before grounding in liquid nitrogen. Then, it was resuspended in an extraction buffer: 0.5 M triethylammonium bicarbonate buffer (TEAB, pH 8.5) with a protease inhibitor cocktail (Roche Ltd. Basel, Switzerland) and transferred into Eppendorf tubes. The suspensions were immersed in a cooled sonication water bath and sonicated for two cycles using a microtip Branson sonifier (Enerson, Danbury, CT, USA). The proteins were collected after centrifuging at 18,000× *g* for 30 min at 4 °C. Samples were then stored at −20 °C for further use.

### 4.4. Protein Digestion, TMT Labeling, and Fractionation

The protein was precipitated at −20 °C overnight with five volumes of pre-chilled acetone. Samples were centrifuged at 15,000 rpm for 15 min at 4 °C and then re-dissolved in 100 µL of 0.5 M TEAB. The protein concentration was measured using the BCA protein assay. Then 100 µg protein of each treatment was taken and incubated with 2 µL of 0.5 M TCEP (Tris(2-Carboxyethyl) Phosphine) at 37 °C for 60 min and subsequently incubated with 4 µL of 1 M iodoacetamide in the dark at room temperature for 40 min. With a ratio of 1:50 (trypsin: protein, *w*/*w*), the protein was digested at 37 °C overnight using a sequence grade modified trypsin (Promega, Madison, WI, USA). Peptides were desalted by C18 ZipTip and then lyophilized by SpeedVac, followed by peptide quantification using a Pierce™ Quantitative Colorimetric Peptide Assay (23275). Peptides were then labelled with a TMT-6 plex Isobaric Mass Tag Labeling Kit (Thermo Fisher Scientific, USA) following the manufacturer’s instruction, pooled, and then lyophilized using a vacuum freeze-drier [37].

### 4.5. Nano UHPLC–MS/MS-Based Protein Identification and Quantitation

The peptides were dissolved in 5% ACN containing 0.5% formic acid and then analyzed by online nanospray LC-MS/MS on Q Exactive™ Lumos coupled to EASY-nLC 1200 system (Thermo Fisher Scientific, USA). Samples were loaded into the analytical system with a trap column (Thermo Fisher Scientific Acclaim PepMap C18, 100 μm × 2 cm) analytical column (Acclaim PepMap C18, 75 μm × 25 cm). The separation procedure was a 60 min gradient from 6% to 30% B (B: 80% ACN containing 0.1% formic acid) with a flow rate of 250 nL/min. The electrospray voltage of 2 kV versus the inlet of the mass spectrometer was applied. The mass spectrometer was run under data-dependent acquisition mode and automatically switched between the MS and MS/MS mode. The parameters were (1) MS: scan range (*m*/*z*) = 375–1600; resolution = 120,000; maximum injection time = 20 ms; dynamic exclusion = 30 s; AGC target = 3 × 10^6^; include charge states = 2–6; (2) HCD-MS/MS: resolution = 30,000; isolation window = 1.2; maximum injection time = 50 ms; AGC target = 2 × 10^5^; collision energy = 32 [38].

### 4.6. Database Search

The raw data on peptides and proteins were converted into .mgf format using Proteome Discoverer (version 1.3.0.339, Thermo Fisher Scientific, Bremen, Germany), and then analyzed by the MaxQuant search engine for protein identification (version 1.5.3.8) [39]. Considering the limited quantity of protein sequences of *C. zofingiensis*, sequence data of *Monoraphidium neglectum* SAG 48.87 and *Chlamydomonas reinhardtii* v5.6 all belong to the Chlorophyceae class. UniProt were used for MS/MS data analysis. The fragment deviation was set at 0.05 Da and a precursor less than 20 ppm with 2 missed cleavages was allowed. Carbamido-methylation (57.02) and TMT 6-plex (K, N-term, 229.16) were selected as fixed modifications, and oxidation (M, 15.99) as a variable modification. The peptides were filtered at the peptide level with a 1% FDR, and one unique peptide at or >95% confidence. All quantified peptides in one protein were combined to calculate the *p*-value (*p* < 0.05, ANOVA). The protein abundance was analyzed using Protein Pilot Descriptive Statistics Template V3.0. Significantly regulated proteins were those showing a 2-fold change. 

### 4.7. Protein Networks and Function Analysis

Hierarchical cluster analysis was carried out to investigate the grouping of samples using the pheatmap package (1.0.12) in R (4.0.2). The volcano plot was performed using the ggplot2 package (3.3.3) in R. Gene Ontology (GO) annotations and the corresponding enzymes commission numbers (ECs) were obtained by Blast2Go analysis. The GO enrichment analysis was assessed using a hypergeometric distribution of the differentially expressed proteins based on the known biological process, cellular component, or biochemical function. Pathway analysis was carried out in KOBAS 3.0 (http://kobas.cbi.pku.edu.cn/), a web server for annotation and identification of enriched pathways. Kyoto Encyclopedia of Genes and Genomes (KEGG) mapping was applied to characterize the metabolic pathways with a hypergeometric distribution model, and the *p* value for each pathway was calculated by taking the total number of mapped genes. A protein–protein interaction network was constructed using STRING (https://www.string-db.org, accessed on 15 June 2022) and visualized by Cytoscape 3.7.1 with *C. reinhardtii* as a reference.

### 4.8. Protein–Protein Interaction Docking by ZDOCK

Molecular docking for the proteins psbB and rbcL was performed using the online docking site ZDOCK server (http://zdock.umassmed.edu/, version 2.3.2, accessed on 7 July 2022) [40]. The top one prediction result was visualized by PyMOL 2.3.4.

## 5. Conclusions

Here, significant algal biomass enhancement (8-fold higher) was obtained in response to a 5 g/L glucose supplement, providing an effective way for future algal commercial or industrial production. Glucose enhances carbohydrate metabolism, resulting in higher cell biomass than that in the control group. Then, TMT proteomic analysis demonstrated the upregulated proteins were mostly associated with cell growth, including DNA-directed DNA polymerase, 40S/60S ribosomal protein, 26S proteasome regulatory subunit, and succinate dehydrogenase, the marker enzyme of the TCA cycle. Moreover, relatively low lipid contents and astaxanthin contents were observed under glucose conditions (7–10 d). Accordingly, the reduced abundance of proteins was mainly seen in the photosynthesis, chloroplast, and thylakoid-associated GO categories, as well as in valine, leucine, and isoleucine biosynthesis, oxocarboxylic acid metabolism, and the pantothenate and CoA biosynthesis KEGG pathways. This study laid a solid basis for better understanding how glucose promotes cell growth of *C. zofingiensis,* which may pave the way for growth trait improvements via genetic engineering of this alga. Of course, there are some limitations to this study: first, the sampling time point for proteomics analysis vary at diverse growth phases; here, only one time point was incorporated. Second, the cell density with/without glucose was significantly different, which may affect the physiological and proteomic responses of the algal cells. Finally, proteome coverage needs to be optimized, as only soluble protein was applied for TMT labeling. Taken together, this proteomic analysis is likely to provide good information about lipid and astaxanthin accumulation in microalgal cells upon glucose supplementation, guiding further research into the investigation of astaxanthin production using genetic or other approaches. It significantly improves our understanding of the molecular mechanisms involved in the tolerance of algae to glucose stress.

## Figures and Tables

**Figure 1 plants-11-01851-f001:**
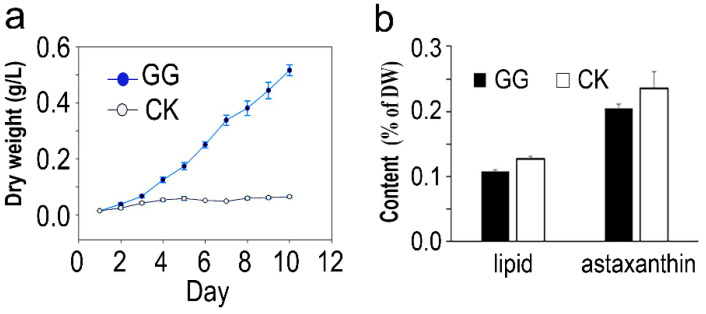
(**a**) Algal dry weight of *Chromochloris zofingiensis* grown under optimal conditions in the medium. (**b**) The lipid and astaxanthin content at Day 10 of *C. zofingiensis* with (GG) and without (CK) glucose addition. Each value represents the mean (±SD) of three biological replicates.

**Figure 2 plants-11-01851-f002:**
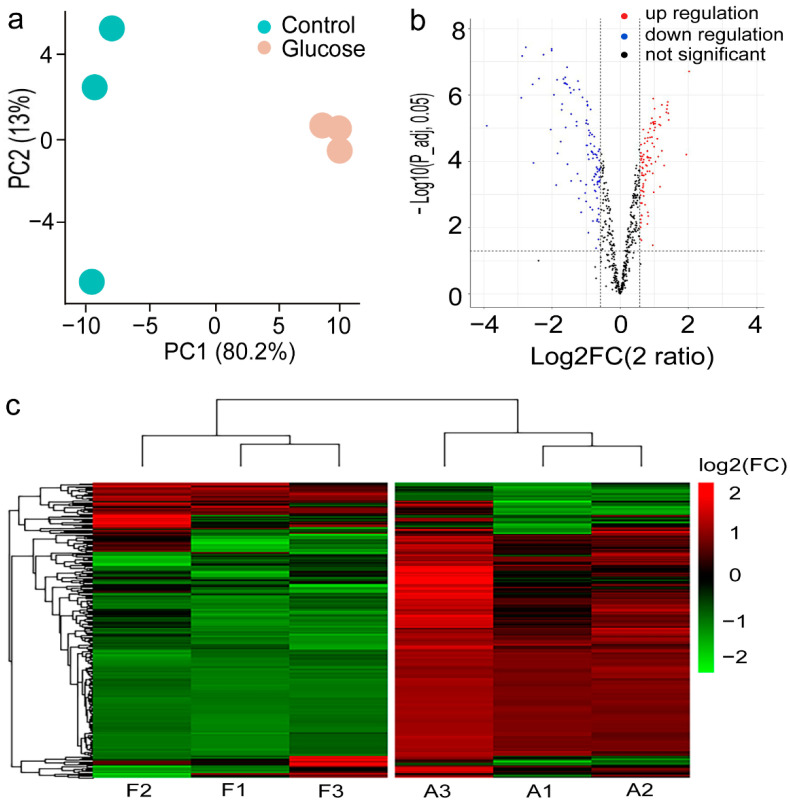
Proteomic analysis of proteins for *Chromochloris zofingiensis* in response to glucose addition by the TMT technique. (**a**) Principal component analysis of the identified proteins. (**b**) Volcano figure of the quantified proteins. The dash line represents the threshold of significant variance (adjusted *p*-value < 0.05), the red dots show the upregulated proteins (log2 (fold change) > 0), the blue one is the downregulated protein (log2 (fold change) < 0), and the black points indicate no significant protein change. (**c**) Hierarchical clustering analysis of all the differentially expressed proteins. F means with glucose, A means without glucose, and 1–3 indicate the biological replicates.

**Figure 3 plants-11-01851-f003:**
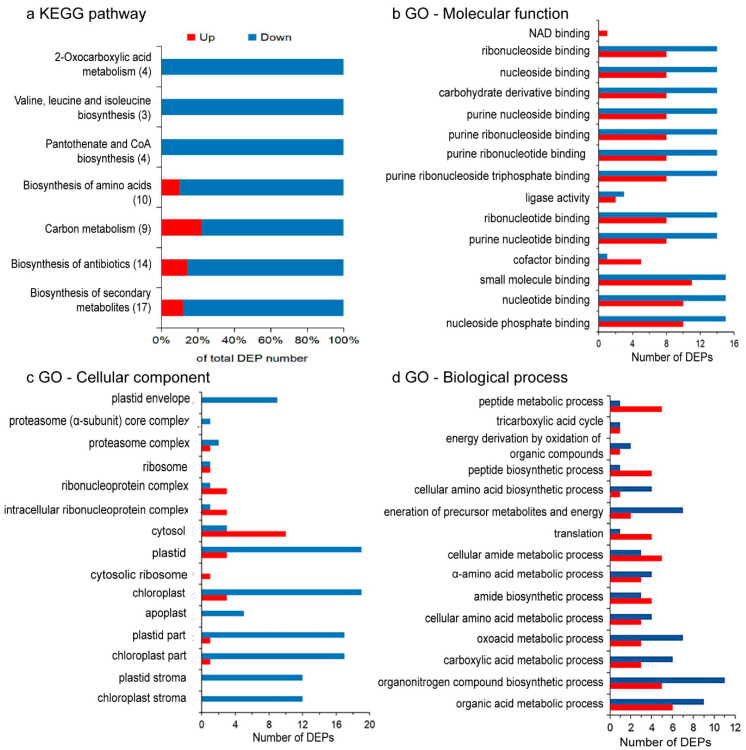
Classification of the differentially expressed proteins (DEPs) in *Chromochloris zofingiensis* with or without glucose. The red and blue colors represent the upregulated and downregulated proteins, respectively. (**a**) Distribution of the DEPs from the crucial KEGG pathways that were significantly enriched; thus, with a *p*-value < 0.05 under the glucose addition condition. Changes are denoted as the percentage of proteins in each pathway. (**b**–**d**) GO classification of DEPs with *p* < 0.01.

**Figure 4 plants-11-01851-f004:**
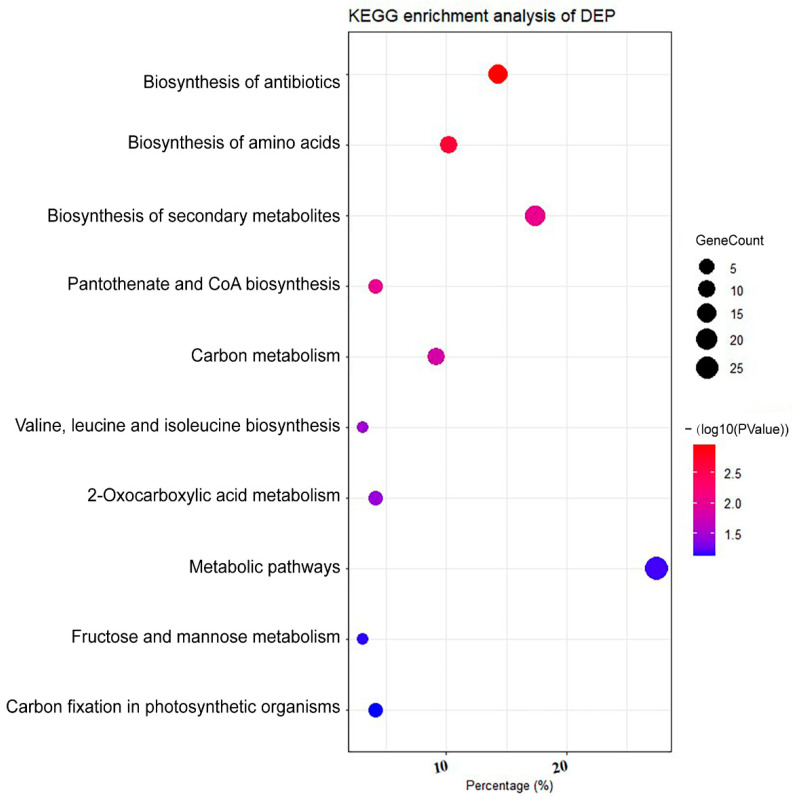
KEGG pathway enrichment result for *Chromochloris zofingiensis* with or without the presence of glucose. The red–purple color scheme represents the *p* value and the circle size indicates the number of analyzed proteins.

**Figure 5 plants-11-01851-f005:**
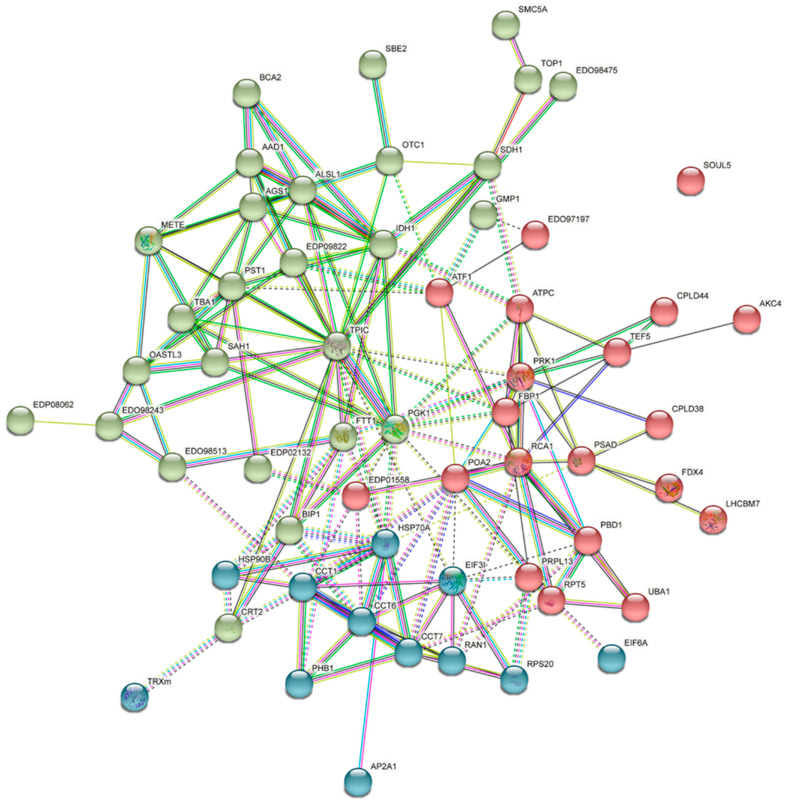
Protein interaction network analyzed in STRING with medium confidence (0.400) and k-means clustering filtered by *Chlamydomonas reinhardtii* (https://www.string-db.org, accessed on 15 June 2022) (see Appendix A).

**Figure 6 plants-11-01851-f006:**
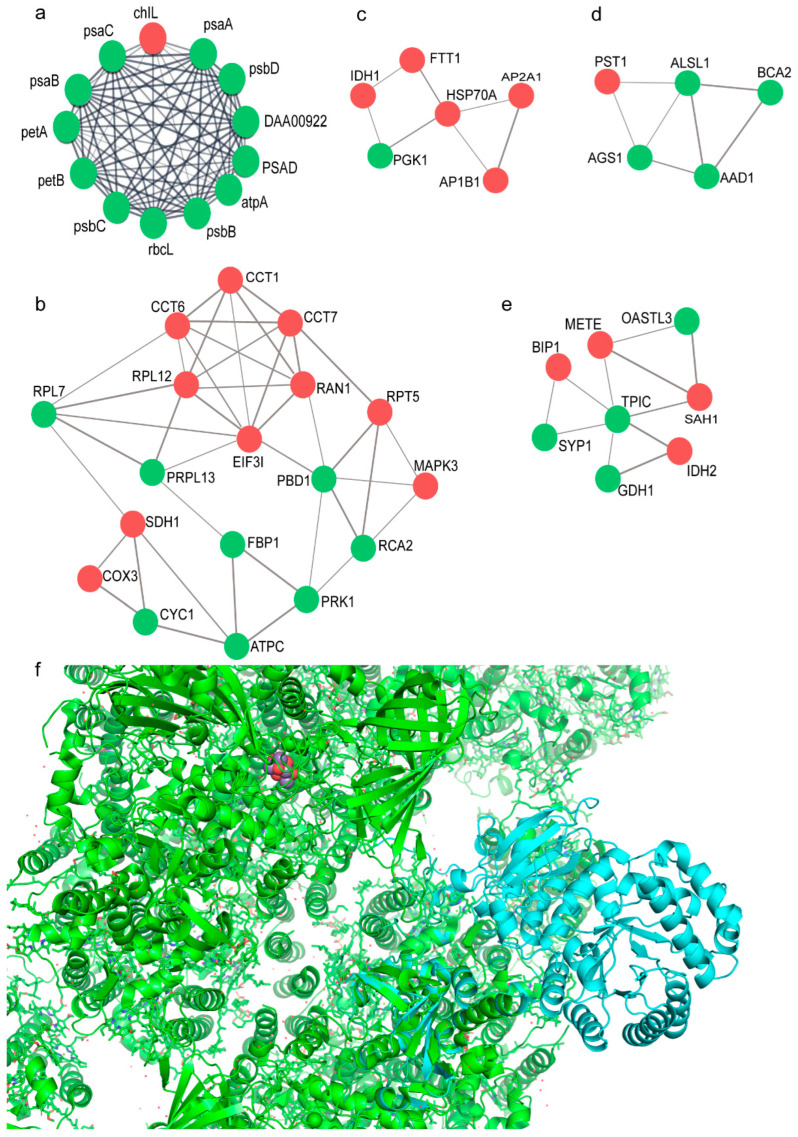
MCODE analysis of the protein–protein interaction network of *Chromochloris zofingiensis* with or without glucose: (**a**) Cluster 1; (**b**) Cluster 2; (**c**) Cluster 3; (**d**) Cluster 4; (**e**) Cluster 5. The node in red color means an upregulated protein; the node in green color represents a downregulated protein. The width of line indicates the interaction strength of the proteins. (**f**) Molecular docking analysis result for psbB (receptor) and rbcL (ligand) by ZDOCK; the receptor is in green and the interaction complex in cyan.

**Table 1 plants-11-01851-t001:** Differentially expressed proteins related to carbohydrate metabolism and growth of *Chromochloris zofingiensis* with or without glucose supplementation.

Gene Name	ID	Annotation	log2 (FC)for G+ vs. G−	Regulation	*p*-Value
OGD1	A8IVG0	2-oxoglutarate dehydrogenase, E1 subunit	0.27	NS	7.18 × 10^−2^
CHLRE_17g713200v5	A0A2K3CPR8	oxoglutarate:malate antiporter	−0.73	down	3.91 × 10^−5^
MNEG_0550	A0A0D2LM39	Putative 2-oxoglutarate/malatecarrier protein	0.93	up	3.36 × 10^−6^
MNEG_6327	A0A0D2MM79	4-hydroxyphenylpyruvate dioxygenase	−0.40	NS	1.14 × 10^−3^
MNEG_9313	A0A0D2JH23	Acyl-carrier-protein desaturase	2.34	up	9.81 × 10^−6^
DGAT1a	A0A411PNH6	Diacylglycerol	1.43	up	5.61 × 10^−7^
CHLRE_03g158900v5	A0A2K3DW88	Dihydrolipoamide acetyltransferase component of pyruvate dehydrogenase complex	0.03	NS	6.08 × 10^−1^
MNEG_1234	A0A0D2MW14	Isocitrate dehydrogenase (NADP(+))	−0.43	NS	5.66 × 10^−4^
CHLRE_02g143250v5	A0A2K3E3Z0	Isocitrate dehydrogenase [NAD] subunit, mitochondrial	0.72	up	7.06 × 10^−6^
IDH3	A8J9S7	Isocitrate dehydrogenase [NADP]	−0.05	NS	5.90 × 10^−1^
IDH1	A8J6V1	Isocitrate dehydrogenase, NAD-dependent	−0.66	down	5.74 × 10^−5^
MNEG_16102	A0A0D2LPD2	Phosphoenolpyruvate carboxylase	−0.05	NS	5.09 × 10^−1^
CHLRE_03g171950v5	A0A2K3DX44	Phosphoenolpyruvate carboxylase	0.08	NS	3.91 × 10^−1^
MNEG_10533	A0A0D2MS98	Phosphopyruvate hydratase	−0.63	down	1.28 × 10^−2^
eno	Q946Z5	Phosphopyruvate hydratase (Fragment)	−0.57	NS	1.65 × 10^−4^
CHLRE_06g258700v5	A0A2K3DMK8	Pyruvate carboxylase	−0.74	down	3.39 × 10^−5^
PYC1	A8HXT4	Pyruvate carboxylase (Fragment)	−0.33	NS	9.08 × 10^−3^
CHLRE_06g258733v5	A0A2K3DMI3	Pyruvate carboxyltransferase domain-containing protein	−0.12	NS	1.89 × 10^−1^
MNEG_8717	A0A0D2KV51	Pyruvate dehydrogenase E1 component subunit alpha	−0.36	NS	1.37 × 10^−3^
CHLRE_02g099850v5	A0A2K3E272	Pyruvate dehydrogenase E1 component subunit alpha	0.02	NS	8.70 × 10^−1^
MNEG_10719	A0A0D2JC20	Pyruvate dehydrogenase E1 component subunit alpha (Fragment)	−1.14	down	2.24 × 10^−4^
MNEG_4864	A0A0D2MRP4	Pyruvate dehydrogenase E1 component subunit beta	−0.27	NS	9.03 × 10^−2^
CHLRE_03g194200v5	A0A2K3DYL5	Pyruvate dehydrogenase E1 component subunit beta	0.13	NS	1.43 × 10^−1^
PDH1a|PDH1b	A8JBC6	Pyruvate dehydrogenase E1 component subunit beta	0.14	NS	6.99 × 10^−2^
MNEG_8504	A0A0D2MZ94	Pyruvate dehydrogenase E2 component (Dihydrolipoamide acetyltransferase)	−0.20	NS	3.24 × 10^−1^
MNEG_13760	A0A0D2J2M4	Pyruvate, phosphate dikinase	−0.73	down	6.85 × 10^−5^
PPD1	A8IC95	Pyruvate, phosphate dikinase	0.11	NS	2.71 × 10^−1^
MNEG_3522	A0A0D2LCG3	26S proteasome regulatory subunit T2	0.853	up	2.85 × 10^−6^
MNEG_4662	A0A0D2MJY5	20S proteasome subunit beta 6	0.620	up	7.11 × 10^−5^
MNEG_6814	A0A0D2MKU1	26S proteasome regulatory subunit T1	0.611	up	5.76 × 10^−5^
MNEG_7470	A0A0D2MIK8	Putative 26S proteasome non-ATPase regulatory subunit 6	0.762	up	4.28 × 10^−5^
MNEG_8800	A0A0D2M741	Proteasome subunit beta	0.800	up	2.14 × 10^−5^
RPT5	A8IIP7	26S proteasome regulatory subunit	0.785	up	1.19 × 10^−5^
PBD1	A8JAI8	Proteasome subunit beta	−0.865	down	4.09 × 10^−6^
POA2	A8JEW4	Proteasome subunit alpha type	−0.585	down	1.42 × 10^−4^

Note: NS—not significant.

**Table 2 plants-11-01851-t002:** Differentially expressed proteins associated with amino acid metabolism of *Chromochloris zofingiensis* with or without glucose.

Gene Name	ID	Annotation	log2 (FC) for G+ vs. G−	Regulation	*p*-Value
CHLREDRAFT_38643	A8J0R6	Alanine-tRNA ligase (Fragment)	0.73	up	7.29 × 10^−^^5^
MNEG_4651	A0A0D2L8Z9	Alanyl-tRNA synthetase	0.64	up	3.26 × 10^−^^5^
CHLRE_06g279150v5	A8J1X8	Aspartyl-tRNA synthetase	0.62	up	3.96 × 10^−^^5^
ATF1	A8IZE7	Glutamine-fructose−6-phosphate transaminase (isomerizing)	0.59	up	1.99 × 10^−^^3^
HemA	Q9FPR7	Glutamyl-tRNA reductase	−0.73	down	3.98 × 10^−4^
CHLRE_16g694850v5	A0A2K3CSB8	Arginine biosynthesis bifunctional protein ArgJ, chloroplastic	−1.06	down	2.48 × 10^−^^6^
MNEG_2778	A0A0D2LET9	Argininosuccinate lyase	−0.72	down	2.12 × 10^−^^5^
MNEG_6059	A0A0D2L3Z0	Glutamate synthase (NADPH/NADH)	−0.84	down	5.27 × 10^−^^5^
MNEG_3551	A0A0D2LCD0	Glutamate synthase (NADPH/NADH)	−0.65	down	8.17 × 10^−4^
MNEG_0007	A0A0D2LNS4	5-methyltetrahydropteroyltriglutamate-homocysteine S-methyltransferase	1.13	up	2.00 × 10^−^^6^
METE	A8JH37	5-methyltetrahydropteroyltriglutamate-homocysteine S-methyltransferase	2.05	up	1.69 × 10^−^^7^
OASTL3	A8IEE5	Cysteine synthase	−0.98	down	1.37 × 10^−^^6^
MNEG_11543	A0A0D2KKW6	Cysteine synthase A	−0.76	down	1.62 × 10^−^^5^
MNEG_10819	A0A0D2M0L3	Cysteine synthase A	−0.62	down	1.57 × 10^−4^
MNEG_3728	A0A0D2K0T8	Glutamine synthetase	−0.64	down	2.26 × 10^−4^
MNEG_7613	A0A0D2MI17	Glutamine synthetase (Fragment)	−0.77	down	9.52 × 10^−^^5^
CHLRE_06g293950v5	A0A2K3DQH9	Serine hydroxymethyltransferase	0.95	up	3.64 × 10^−^^6^
PST1	A8IH03	Phosphoserine aminotransferase	1.14	up	4.96 × 10^−^^6^
MNEG_16458	A0A0D2LNB1	Serine/threonine-protein phosphatase	−1.24	down	6.35 × 10^−^^7^
AAD1	A8IX80	Acetohydroxyacid dehydratase	−2.28	down	8.58 × 10^−^^7^
ALSL1	A8J1U3	Acetolactate synthase, large subunit	−0.84	down	1.81 × 10^−4^
MNEG_15388	A0A0D2MB71	Isoleucyl-tRNA synthetase	0.65	up	2.83 × 10^−^^5^
BCA2	A8I5J8	Branched-chain-amino-acid aminotransferase	−0.66	down	1.57 × 10^−4^

**Table 3 plants-11-01851-t003:** Differentially expressed proteins involved in energy of *Chromochloris zofingiensis* with or without glucose supplementation.

Gene Name	ID	Annotation	log2 (FC) for G+ vs. G−	Regulation	*p*-Value
ycf3	A0A140HA77	Photosystem I assembly protein Ycf3	−0.31	NS	3.37 × 10^−^^3^
psaC	A0A140HA43	Photosystem I iron-sulfur center	−1.36	down	2.14 × 10^−^^6^
psaA	A0A140HA40	Photosystem I P700 chlorophyll a apoprotein A1	−2.29	down	4.13 × 10^−8^
psaB	A0A140HA64	Photosystem I P700 chlorophyll a apoprotein A2	−1.82	down	1.41 × 10^−7^
PsaD	Q5NKW4	Photosystem I reaction center subunit II, 20 kDa	−1.44	down	2.86 × 10^−4^
psaN	Q9AXJ2	Photosystem I reaction center subunit N	−2.04	down	1.06 × 10^−^^5^
psbC	A0A140HA26	Photosystem II CP43 reaction center protein	−1.64	down	2.98 × 10^−7^
psbB	A0A140HA55	Photosystem II CP47 reaction center protein	−1.49	down	3.47 × 10^−7^
psbD	A0A140HA41	Photosystem II D2 protein	−1.58	down	3.30 × 10^−7^
psbA	A0A140HA23	Photosystem II protein D1	−1.48	down	3.43 × 10^−7^
MNEG_8562	A0A0D2M7P8	Photosystem II stability/assembly factor (Fragment)	−1.37	down	2.00 × 10^−7^
PsbP domain-containing protein	A0A2K3D661	Photosystem II PsbP domain-containing protein	−1.55	down	1.00 × 10^−4^
rbcL	A0A140HA49	Ribulose bisphosphate carboxylase large chain	−2.05	down	4.65 × 10^−8^
rbcL.1	A0A218N8A3	Ribulose bisphosphate carboxylase large chain	−0.23	NS	2.00 × 10^−^^1^
rbcL.3	A0A517BB24	Ribulose bisphosphate carboxylase large chain (Fragment)	−1.36	down	4.31 × 10^−4^
rbcL.4	Q3HTJ4	Ribulose bisphosphate carboxylase large chain (Fragment)	−2.29	down	3.03 × 10^−4^
RBCS	M4QL06	Ribulose bisphosphate carboxylase small chain	−0.97	down	7.98 × 10^−4^
MNEG_12441	A0A0D2KIA7	Ribulose bisphosphate carboxylase/oxygenaseactivase	−1.04	down	3.04 × 10^−^^3^
SDH1	A8HP06	Succinate dehydrogenase [ubiquinone] flavoprotein subunit, mitochondrial	0.83	up	5.93 × 10^−^^6^
MNEG_7567	A0A0D2N2H0	Thylakoid formation protein 1	−0.39	NS	1.03 × 10^−^^1^
MNEG_2022	A0A0D2MTM8	Thylakoid lumenal 17.4 kDa protein, chloroplastic	−1.50	down	4.25 × 10^−^^5^
CPLD44	A8J6G0	Thylakoid lumenal protein	−1.16	down	8.51 × 10^−^^6^
MNEG_1868	A0A0D2K714	Pyruvate kinase	0.19	NS	2.96 × 10^−2^
MNEG_11538	A0A0D2LYE6	Pyruvate kinase	−0.19	NS	3.69 × 10^−2^
PYK1	A8IVR6	Pyruvate kinase	0.03	NS	8.10 × 10^−^^1^
PYK2	A8J214	Pyruvate kinase	0.56	NS	1.13 × 10^−4^

Note: NS—not significant.

## Data Availability

The LC-MS raw data were uploaded to ProteomeXchange with PXD034488 as the original dataset.

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
