# Peer review of "Quantitative Proteomics of Chromochloris zofingiensis Reveals the Key Proteins Involved in Cell Growth and Bioactive Compound Biosynthesis"

_plants, 2022, doi:10.3390/plants11141851_

Round 1

Reviewer 1 Report

The authors carried out a study entitled "Quantitative proteomics of Chromochloris zofingiensis reveals key protein involved in cell growth and bioactive compound biosynthesis"

In my opinion the manuscript is well written and the results are solid, after minor revisions the editor can accept the work for publication.

I recommend that authors add a molecular modeling study using docking that can significantly improve the quality of the manuscript, in addition to showing molecular interactions

Author Response

Response: Thanks for the positive comments. Protein-protein interaction analysis was applied by taking psaB and rbcL as an example using online docking site ZDOCK server. Related results were included in the manuscript. Results section in lines 197-197, 200, 204-205. Methods section in lines 418-421.

Reviewer 2 Report

I commend the authors: Wen Qiua, Rongfeng Chen, Xianxian Wang , Junying Liug, and Weiguang Lv of the manuscript titled “Quantitative proteomics of Chromochloris zofingiensis reveals key protein involved in cell growth and bioactive compound biosynthesis” for their work on proteomics response on astaxanthin biosynthesis of Chromochloris zofingiensis algae.

Before this manuscript is published, there are several things need to be addressed or corrected:

1-       In the introduction, some linguistics need to be improved such as sentence lines 66-68.

2-      In the results

-          Figure 2 need to be enlarged 2 times at least

-          Figure 3 and 4 need to be enlarged, enhanced because its blurring.

3-       

4-      In the discussion

-          The discussion need to be enhanced and to be more concise, its long and need to be shortened, especially from lines 264-319.

5-      the materials and methods is well written.

6-      The conclusion, add the prospect of the work.  Add the limitation of the study as well.

I give you major revision.

7-      The references should be reduced because this is not a review article.

Author Response

Response to Reviewer 2 Comments

Point 1-      In the introduction, some linguistics need to be improved such as sentence lines 66-68.

Response: The sentence was improved.

Development on techniques enable researchers to exploit key genes regulating astaxanthin production [9]. It also accelerates to elucidate mechanisms modulating lipid/astaxanthin accumulation in response to external stimuli [11].

Point  2-      In the results

-          Figure 2 need to be enlarged 2 times at least

Response: Figure 2 has been enlarged as required.

-          Figure 3 and 4 need to be enlarged, enhanced because its blurring.

Point  3-       Response: Figure 3 and 4 have been enlarged and enhanced as required.

Point  4-      In the discussion

-          The discussion need to be enhanced and to be more concise, its long and need to be shortened, especially from lines 270-320.

Response: Thanks. The mentioned discussion was modified to be more concise for lines 268-318.

Glucose cultivation causes the downregulation of 11 proteins involved in photosynthesis (photosystem I and II, detailed in Table S3), implying that photosynthesis and electron transport is largely inhibited in Chlorella cells with glucose supplement. It agrees well with the report for Chlorella protothecoides sp. 0710 [27] and C. zofingiensis SAG 211-14 [21, 22], for which heterotrophic culture with glucose cause almost completely degradation of enzymes associated with photosynthesis [13]. It also consists with transcriptome result from Roth et al [21, 22], who observed the absence of photosynthetic activity and the significant decreased expressions of gene for photosystem I and II in C. zofingiensis SAG 211-14 with the presence of glucose through RNA-seq. Indeed, the availability of glucose stops the necessity of algae to obtain organic carbon through photosynthesis process [27] and trigger the turns off of photosynthesis, degrade the photosynthetic apparatus, and reduce thylakoid membranes under light condition [22]. On the other hand, glucose may alter algal cellular lipid composition or cell structure e.g., decrease in chlorophyll content and chloroplast degradation, where astaxanthin biosynthesis process occurs [10].

Moreover, we detected the reduced abundance of other photosynthesis associated proteins including phosphoribulokinase, acetyl-CoA carboxylase, and pyruvate dehydrogenase complex SBPase in C. zofingiensis SAG 211-14. It is unexpected, because feeding glucose may recalibrate cell metabolism towards downstream intermediates and lipid accumulation in C. protothecoides [13]: phosphoribulokinase catalyzes the production of ribulose 1,5-bisphosphate, and the substrate functions to capture CO2 in photosynthesis and fatty acid synthesis in photosynthetic organisms [23]. As for acetyl-CoA carboxylase, it catalyzes the first and rate-limiting step for fatty acid synthesis pathway, while pyruvate dehydrogenase complex is involved in acetyl-CoA formation. Overexpression of SBPase in plants [24], microalgae Dunaliella bardawil [28] and C. reinhardtii [29] leads to significant increase in photosynthesis, addressing the importance of these proteins.   

Furthermore, glucose supplement might reduce the carotenoid biosynthesis sites because of smaller chloroplasts under glucose conditions (smaller cell size) and thus account for the lower astaxanthin content [10]. Glucose may directly regulate astaxanthin accumulation through modulating the expression of key genes modulate astaxanthin biosynthesis such as β-carotenoid ketolase (BKT) and β-carotenoid hydroxylase (CHYb) genes of C. zofingiensis [30]. It may affect the transcription of BKT and CHYb genes through affecting de novo protein synthesis [30], because increasing glucose supply decreased protein content [15]. Although key proteins involved in astaxanthin production (BKT and CHYb) were not detected, inhibition response of fatty acid production with glucose supplement (Figure 2) and the downregulation of photosynthesis associated proteins (Table 3) verified the proposed linkage between photosynthesis and lipid production in microalgae Eutreptiella sp. [31]. In addition, glucose exposure enhanced the expression of PDAT gene, which may promote the chloroplast decomposition, and thus reduce the available synthetic sites for astaxanthin biosynthesis in C. zofingiensis [10]. The balance between astaxanthin production, cell growth and biomass accumulation could be modulated by the supplements with glucose (C/N ratio) for large-scale cultivation [15, 21]. Accordingly, this study provides hints for its biotechnological modification: carbon sources like glucose are used to provide more energy for higher growth rate as well as for respiration, the cellular physiology and morphology would be changed via affecting metabolic pathways of carbon assimilation and allocation [27].

The network showed the expression of transcription and translation relating proteins reflecting the significant change in C. zofingiensis in response to the glucose-induced condition [38]. Abundances altered proteins are likely to provide new insights about lipid accumulation in microalgal cells after glucose supplement. Much work remains in better understanding the differences in regulation of chloroplast structure and carbon flow upon the glucose supplies to algal culture [32].

Point  5-      the materials and methods is well written.

Response: Thanks for the positive comments.

Point  6-      The conclusion, add the prospect of the work.  Add the limitation of the study as well.

Response: The prospect and limitation of the study was added in the Conclusion section in lines 424-426, 435-442:

Here, significant algal biomass enhancement (8-fold higher) was obtained in re-sponse to 5 g/L glucose supplement, providing an effective way for future algal commer-cial or industrial production. 

This study laid a solid basis for better understanding glucose promotes cell growth of C. zofingiensis, which may pave the way for growth trait associated improvement via genetic engineering in this alga. Of course, there are some limitations in this study: first, sampling time point for proteomics analysis varied at diverse growth phase, here, only one time point is incorporated. Second, cell density under with/without glucose conditions is significantly different, which may affect the physiological and proteomic responses of algal cell.

Point  7-      The references should be reduced because this is not a review article.

Response: The references have been reduced from 47 to 39 as required.

Reviewer 3 Report

The main goal of this research is very important. This study provides some new information about proteins involved in bioactive compound biosynthesis.

Some minor mistakes:

Introduction line 61 Chlorella vulgaris should be italics.

In tables, the font could be smaller to make it easier to read.

Line 167 CO2 fixation should be corrected into CO2 fixation (and some similar mistakes like in line 248,285)

The description for Figure 5 should be below the figure.

Lines 201-203- This sentence is a repetition of information contained in the chapter Results. Here there should only be a discussion of the results obtained.

Line 291 C. reinhardtii should be italics.

Line 326 30 µE m-2 s-1 the record of the unit should be corrected.

Line 355 ddH2O should be corrected into ddH2O

Author Response

Point 1  Introduction line 61 Chlorella vulgaris should be italics.

Response: Sorry for the negligence. Chlorella vulgaris is changed to italics.

Point 2  In tables, the font could be smaller to make it easier to read.

Response: Font size has been changed from 11 to 9 for all tables to make it read easier.

Point 3  Line 167 CO2 fixation should be corrected into CO2 fixation (and some similar mistakes like in line 248,285)

Response: Thanks. The written of CO2 fixation has changed into CO2 fixation.

Point 4  The description for Figure 5 should be below the figure.

Response: Thanks. The position of figure legend for Figure 5 was changed as required.

Point 5  Lines 201-203- This sentence is a repetition of information contained in the chapter Results. Here there should only be a discussion of the results obtained.

Response: Thanks. The relevant sentence was modified in lines 213-214.

Point 6  Line 291 C. reinhardtii should be italics.

Response: C. reinhardtii is changed to italics.

Point 7  Line 326 30 µE m-2 s-1 the record of the unit should be corrected.

Response: The unit is corrected for light intensity 30 µE m-2 s-1 .

Point 8  Line 355 ddH2O should be corrected into ddH2O

Response: Thanks, ddH2O has changed to ddH2O.

Round 2

Reviewer 2 Report

Accepted for me